# Unveiling Transboundary Challenges in River Flood Risk Management: Learning from the Ciliwung River Basin

Harkunti Pertiwi Rahayu[1], Khonsa Indana Zulfa[2], Dewi Nurhasanah[1], Richard Haigh[3], Dilanthi Amaratunga[3], In In Wahdiny[1]

[1]School of Architecture, Planning and Policy Development, Bandung Institute of Technology, Bandung, 40132, Indonesia
[2]Institute for Risk and Disaster Reduction, University College London, WC1E 6BT, UK
[3]Global Disaster Resilience Centre, University of Huddersfield, Huddersfield, HD1 3DH, UK

*Correspondence to*: Harkunti Pertiwi Rahayu (harkunti@pl.itb.ac.id)

**Abstract**

Due to massive development in urban and surrounding rural areas, the Greater Jakarta Metropolitan has experienced a dramatic increase in the impacted area and amount of economic loss from annual flooding in the Ciliwung River Basin. As the longest river basin crossing many cities and regencies, the complexity of Ciliwung River flood management has been driven by many driving factors triggered not only by natural, physical, and socio-economic factors, but also by transboundary issues and power sharing. Previous studies addressed these flood drivers but have tended to overlook transboundary issues and power sharing. To tackle future flood events, this paper attempts to unveil transboundary issues and power sharing for river flood and water resource management. In this study, 13 significant transboundary flood drivers were identified from literature and practices. Using MICMAC, a power-dependency model, this study is able to further recognize strategic key flood drivers from key stakeholders' perspectives obtained from in-depth interviews and FGDs. Findings of the study show that a lack of control of spatial plans and weak stakeholder coordination-cooperation are found to be the critically important drivers to prioritize, since they have strong impact to all other drivers. Finally, this study proposes that a national-level development control regulation and an acting commission are established as a priority action for transboundary flood risk management in the Ciliwung River Basin. These findings contribute to the literature on governance of flood risk management by emphasizing the need for a coordinated and integrated approach to mitigate flood risks that extend beyond administrative boundaries, enhancing the overall resilience and sustainability.

*Keywords: Flood risk driver, Key flood risk driver, Ciliwung river basin, transboundary flood risk management, MICMAC analysis, Greater Jakarta Flood, Spatial Plan, Stakeholder Coordination and Collaboration*

## 1. Introduction

Due to massive urbanization and development, urban growth often forces the spatial expansion of cities to form an urban agglomeration (metropolitan), which is not only strengthening regional economic development but also reveals the complexity of effective governance in transboundary river flood management. Transboundary rivers basins are those that crossing political and administrative jurisdiction borders between countries or cities/regencies. In the past two decades, urban agglomerations are becoming a vital geographic unit for nations to sustain economic growth. The 40 largest urban agglomerations cover 18% of the world population, 66% of global economic activities, and approximately 85% of technological innovations (UN Habitat, 2016) By 2022, about 56% of the world population lives in urban area and expected to increase to 68% by 2050 (United Nations, 2022). However, the integration of multiple cities in urban agglomerations not only creates a complex, dynamic, and huge system (Müller, 2013), but also blurs the boundaries between cities and its peripheral regions, as well as its administrative system that will impact the governance of transboundary river flood risk management.

Several works have attempted to solve challenges related with the governance of transboundary river management. The use of ecology-water-energy-food nexus' evolutionary game model to understand and mapping the complex and dynamic interrelationship among transboundary challenges (Guo et al., 2024), an advancement of classical game theory used to solve the non-dynamic transboundary challenges and actors (Zhang et al., 2024). However policy makers and river managements authorities have often been exposed to the challenges of managing complex spatial and temporal cause effect relationships when attempting to reduce river flood risk, and also faced conflicting transboundary interests (Lorenz et al., 2001). Further, harmonizing the horizontal and vertical coordination among stakeholders/actors in integrated flood risk management plays important role in reducing the exposure risk (Clegg et al., 2023; Lorenz et al., 2001).

Meanwhile, extensive growth of urbanization severely impacted on environmental challenges, i.e. climate change, environmental degradation, and disaster (Lin et al., 2020), with flood hazard as the most significantly destructive and frequent threat to built environment and human being globally. For example, Indonesian natural disaster profile from 1815 to 2023 shows that flood is the highest frequency events with 13,915 events with 30,671 fatalities (National Disaster Management Agency - BNPB), 2024). The occurrence of urban flood in China often affected extensive regions, such as urban agglomeration, revealing risk associated with complex environmental attributes, social economic attributes, physical attributes, and economic structures (Y. Wang et al., 2023). In fact, many coastal urban agglomerations in developing countries, such as in Indonesia and Chile, are threatened by a range of hydrometeorological hazards, climate induced extreme events, climate change impacts, and anthropogenic threats which lead to the increase of urban flood - a combination of fluvial flood, pluvial flood, and coastal flood (Ariyani et al., 2022; Khoirunisa & Yuwono, 2023; Müller, 2013; Rahayu & Nasu, 2010; Texier, 2008a).

The complexity of urban flood is triggered by risk factors and its collateral, which is in this study called as flood risk challenges. For example, the increase of population growth will lead to the increase of population density, demographic profiles, assets to support growth, land use change of the river basin, land subsidence due to the excessive use of ground water, low awareness, slump area, and waste/trash which decreased the river and drainage capacity (Rahayu, 2022a). Several physical and natural factors significantly influenced the urban flood are tidal surge, extreme intensity, river physiography, erosion, sedimentation, and inappropriate flood control structure (Rahayu, 2022b).

Several quantitative studies on flood risks assessment at the scale of urban agglomerations revealed a significant gap between subregions within large-scale regions from the perspectives of spatial temporal issues, i.e. land cover characteristics, topography, and water networks (Liu et al., 2021). However many of them only focus at small scale of urban agglomeration, such as the city level (Pandey et al., 2023; Wang et al., 2022, 2023). Meanwhile, a qualitative study using absorptive capacity, adaptive capacity, and transformative capacity approach has attempted to identify transboundary flood resilient challenges, for the case of Narayani and Mahakali basins in Nepal (Pandey, 2023). They are early warning system, communication system, disaster risk reduction institutions, and community capitals-safety nets as absorptive capacity; meanwhile crops-livelihoods, infrastructures, and income diversification as adaptive capacity; while public, private, and civil society initiatives and partnerships as transformative capacity.

From the governance perspective, a multi-level transboundary authority of flood risk management shows several challenges, such as silo jurisdiction which is silo in coordination among transboundary administrative regions due to ego sectoral, hegemony of actors and dynamic perceived power between various actors (Polese et al., 2024). A study using Flood Risk Management FRM approach also found several multiple transboundary challenges in reducing the flood impacts, i.e. land-use change, climate change, infrastructure management, and institutional capacity (Mehta & Warner, 2022). Thus, harmonizing or synergizing those challenges arising from the performing of transboundary flood risk reduction and management is critically necessary.

Until now, the most critical issue for transboundary flood risk management has not been deeply studied, i.e. prioritizing the transboundary flood risk challenges for the purpose of disaster risk reduction intervention actions. Prioritizing challenges is very important in the development of collective action plan for transboundary flood risk management both horizontally among

the same level of local or provincial administrative jurisdiction and vertically among local with provincial and/or national administrative jurisdictions. To untangle the complexity of the river flood risk, we must "dig at the roots instead of just hacking at the leaves", we must fully understand "What are the main transboundary challenges to managing river flood?" and "What are the most strategic intervention for reducing the transboundary flood challenges?". Up until recently, these questions remain unsolved.

Therefore, this study aims to recognize the actual condition of the challenges of transboundary flood risk reduction called driving variables, and then to develop a model for prioritizing the transboundary challenges, which is importantly needed to structure collective transboundary flood risk reduction actions. This will be achieved using Matriced' Impacts Croisés Appliquée á un Classement (MICMAC) analysis, learning from Ciliwung River Basin. Holistic data acquisition of transboundary challenges of flood risk obtained through documents reviews, in-depth interviews, observation survey, and FGDs. The resulting MICMAC model maps the interrelationships among those flood driving variables using the degree of power and dependency criteria. This can enable understanding how cross-border cooperation and local/regional/national level policies can be used to synergize and structure those collective action to reduce and manage flood risk effectively.

In general, recognizing the transboundary flood risk drivers and the priority strategic countermeasures to reduce and manage the key flood drivers, this study will enrich the area of flood disaster risk management, water resource management, environmental disaster science and governance of transboundary river management. This study is relevant for areas of research involving the management of shared water resources, the impact of regional development on flood risk, and strategies to reduce economic losses from flooding. With research emphasises on the cross-border administrative characteristics such as in the Ciliwung watershed, this study offers a unique perspective on the challenges and solutions associated with flood risk management in a region involving several administrative jurisdiction regions.

This paper is divided into three main parts. The first, materials and methods, in-depth literature and document reviews for recognizing transboundary flood risk drivers/challenges and the governance of transboundary flood reduction and management. Then using MICMAC model analysis for the prioritizing transboundary challenges of flood risk reduction and management using power dependence relationship. The second part exemplifies these theoretical considerations by using Ciliwung river basin in Metropolitan Jakarta as the case study. The final discussion recaps the key challenges of transboundary flood risk reduction and management as the findings of MICMAC analysis model, as well as its interaction in the disaster risk reduction collective actions.

## 2. Materials and Methods

### 2.1 Flood Risk Drivers Framework

Globally, floods have emerged as one of society's most dangerous risks (Beese et al., 1999), and the most frequent disaster faced by many urban areas in Indonesia (Rahayu & Nasu, 2010). There has been a significant increase in damages caused by catastrophic urban flooding over the past 50 years (Munich Re Group, 2004). The flood risk is created by the combination of flood hazards and vulnerabilities (Beese et al., 1999; UNISDR, 2009), and it refers to the likelihood and exposure of elements to flood hazards.

As discussed in the introduction, many approaches have been used to define a flood risk drivers, mainly they used a hazard, vulnerability and capacity HVC approach (Ariyani et al., 2022; Collalti et al., 2024; Müller, 2013), using an exposure risk approach (Wang et al., 2023), using a resilient component approach, i.e. absorptive capacity, adaptive capacity and transformative capacity (Pandey et al., 2023), using the integration of a CCA and DRR approach (Booth et al., 2020), and using Integrated Water Resource Management – IWRM (Zeitoun et al., 2013).

To have better understanding of flood risk reduction intervention action, the flood risk drivers are defined as an event that can change the condition of a flooding system and are characterized by using the source-pathway-receptor (SPR) paradigm (E.

Evans et al., 2006). A flood source is determined as any event or condition that may cause flooding due to meteorological conditions (e.g., extreme rainfall, sea level rise), while pathways is a mechanism to transfer floodwaters to the locations where they may impact receptors, and receptors are people and built environments that may be impacted by flooding.

Based on the previous works (E. P. Evans et al., 2008; O'Donnell & Thorne, 2020), a source-pathway-receptor (SPR) river flood driver framework is shown in **Table 1.** There are five drivers identified flood source, such as temperature, precipitation, sea-level rise, storm surges, and waves. Nine flood drivers are categorized as flood pathway, they are river morphology, river vegetation, sediment supply, groundwater flooding, sewer conveyance, urbanization, land-use change, environmental regulation, and stakeholder behaviours. The other five drivers are defined as flood receptor, i.e., urban impact, buildings, infrastructure impact, economic impact, and social impact.

**Table 1: Source-pathway-receptor (SPR) Framework for Transboundary River Basin**

| Group | Flood Drivers |
|---|---|
| **Source** | Temperature |
| | Precipitation |
| | Sea-level rise |
| | Storm surges |
| | Waves |
| **Pathway** | River morphology |
| | River vegetation |
| | Sediment supply |
| | Groundwater flooding |
| | Sewer conveyance |
| | Urbanization |
| | Land-use change |
| | Environmental regulation |
| | Stakeholder behaviours |
| **Receptor** | Urban impact |
| | Buildings |
| | Infrastructure impact |
| | Economic impact |
| | Social impact |

Extreme precipitation is known to significantly affect floods in many metropolitan areas of Indonesia, including floods in Greater Jakarta (Mishra et al., 2018). Sudden changes of extreme precipitation in a short duration, which leads to an increase of water volume, intensity, duration, and location, may cause severe flooding (O'Donnell & Thorne, 2020). The issue of transboundary river flood is that flood occur not only by upstream precipitation but also due to downstream rainfall. In many cases, according to rainfall spatial distribution data, most of the metropolitan floods, including Greater Jakarta floods, were caused by evenly distributed rainfall along the Ciliwung River Basin (Farid et al., 2021).

It is understood that fluvial floods occur when rivers do not have sufficient capacity to pass flow rates from upstream to downstream (Asdak, 1995). The narrowing of the river capacity is mainly due to sedimentation and waste, as well as the construction of settlements on uncontrolled riverbanks. However, reducing river sedimentation have been done by lowering the river flow rate by constructing man-made lake, dam, and canal. For example, several man-made lake and dam have been constructed in upstream area of Ciliwung river, such as Bogor Regency and Depok City area, to control flood peak discharge in mid-stream and downstream areas of the Ciliwung River Basin (Nugraheni et al., 2020).

To have better visualisation, this study has preliminarily attempted to describe the S-P-R flood risk driver framework in fish
the bone diagram, see **Figure 1** (Rahayu et al (2022). This diagram was developed by integrating hazard, vulnerability, and
capacity approach into SPR. The source is defined as flood hazard, i.e. extreme precipitation, climate change impact, pathway
is defined as drainage and absorption capacity, while receptor is defined as social and physical vulnerabilities, as well as
community and government capacities.


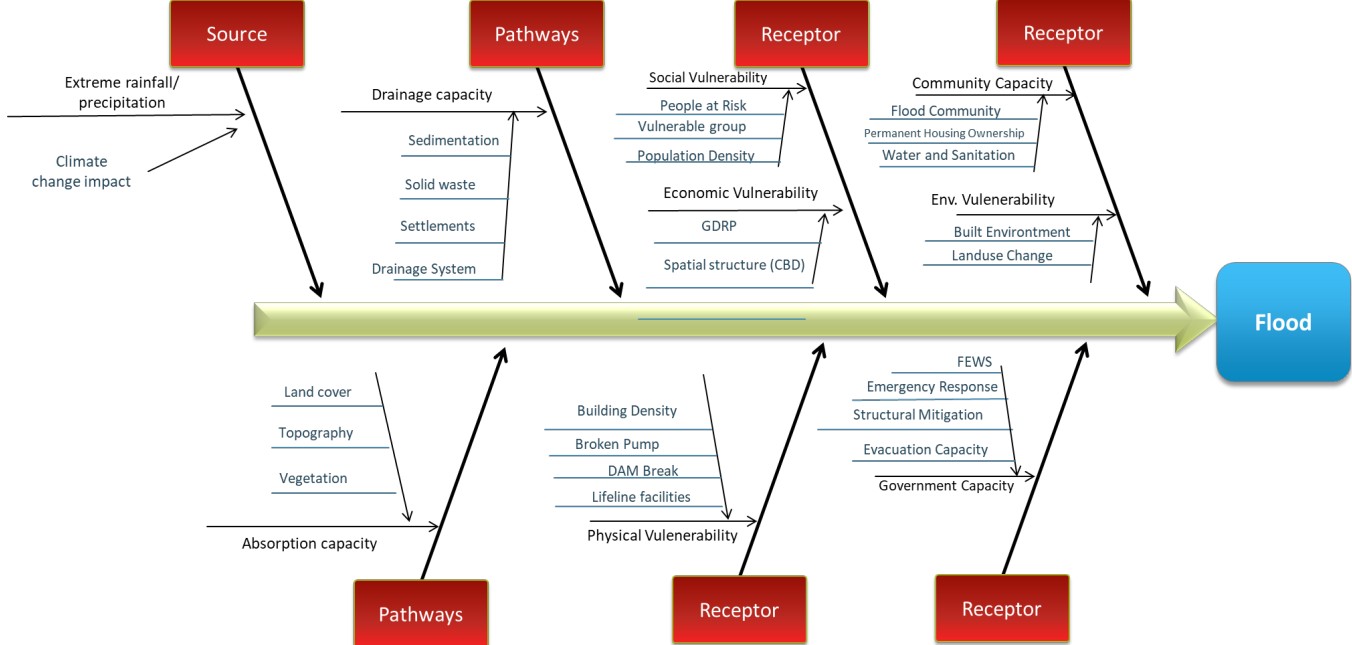

**Figure 1: Key Flood Drivers**

## 2.2 MICMAC

Prioritizing the challenges of transboundary river flood risk can be done by many approaches. The 20/80 Pareto principles
commonly used classification, i.e. 20% of the population possesses 80% of the wealth. The prioritizing using the Pareto
chart, i.e. a simply block diagram displaying relative degree of important of several different attributes in descending order.
Then the top 20% population is believed to have influence on the 80% population of attributes (Grosfeld-Nir et al., 2007).
Meanwhile, Pareto efficiency has been used for climate trade policies and border tax adjustments (Keen & Kotsogiannis,
2014). The weakness of this approach is when the Pareto chart of the data having almost flat distribution. This approach will
be very useful if we have a steep descendent curve, where 20% - 80% can be allocated based on the frequency distribution.
Others used power-interest matrix, a new approach for prioritizing role-based stakeholders' engagement (Hourton,
2014, and Dutta 2020), mapping the variables in four quadrants. First quadrant is the key players with high power and
high interest. Second quadrant is the keep informed with low power but high interest. Third quadrant is the minimal
effort with low power and low interest. The fourth quadrant is the keep satisfied with high power and low interest.

collective action of disaster risk reduction intervention among transboundary stakeholders

Several studies have used power-dependence approach, a very simple and useful approach for classification and structuring
the priority of flood drivers. To identify driving power (influential) and dependence power (influenced), a quantitative method
called Matriced' Impacts Croisés Appliquée á un Classement (MICMAC) is a powerful tool. Both Attri et al. (2013) and Saxena
et al. (1990) are agreed that MICMAC analysis is a significant tool for an in-depth analysis of a system. The method defines

the level of power and dependency by analysing the interrelation among the drivers. MICMAC analysis is carried out to scrutinize the impact of driving power and dependency power of the factors (Ansari et al., 2013). Putting the driving power along X-axis and dependency power along Y axis, the factors are classified into four quadrants (group), i.e. independent, linkage, autonomous, and dependent groups(Duperrin & Godet, 1973; Jharkharia & Shankar, 2005).

The independent group consists of drivers with strong driving power but low dependence power. These driving factors, often called key driver variables, are rarely influenced by other drivers, but strongly influence other groups. While the linkage group consists of the driving factors characterised by both high driving and high dependency power. Factors within the linkage group both influence other groups as well as being influenced by them. This group, called relay variables, represents factors that have strong interconnections and mutual dependencies with factors in other quadrants. Meanwhile, the autonomous group, often 185 called autonomous variables, consists of driving factors characterised by extremely low driving power while also having low dependency power on factors in other groups. They are self-contained and do not significantly drive or depend on other quadrants. Then, the dependent group, called as dependent variables, consists of driving factors characterised by low driving ability but high reliance power on other groups. These drivers rely heavily on other drivers, and any action taken by other drivers will have an impact on the dependent drivers.

In disaster risk reduction, MICMAC analysis was previously used to discover important factors of resilient humanitarian supply chain that emerge during post-disasters (Singh et al., 2018). The study's findings will help government agencies and policymakers make proper strategic decisions to increase resilience. Further, it assists emerging countries in minimizing massive losses and improving economic growth for the benefit of society. However, up to these days, there is only few published studies in discovering key flood drivers using the MICMAC method. For example (Ariyani et al., 2022) have used 195 MICMAC model to identify the contributing factors influencing flood disaster in Ciliwung watershed using influence-dependence criteria. The factors were identified using hazard vulnerability capacity framework; however the targeted respondent was very limited and not comprehensively represented by flood direct related institutions of transboundary administrative jurisdiction. Just showing 1 Jakarta Provincial Government, two national institutions and the rest are experts. Meanwhile, Sharma 2019, attempted to conduct identification and prioritization of flood conditioning factors using ISM and 200 MICMAC technique. About 11 flood conditioning factors are identified based on literature review and Deplhi Method for expert judgment. The credibility information of the study will depend on the selected experts.

The improve the deficiency of the above study, this study use the MICMAC analysis model in prioritizing the flood drivers to obtain the key flood drivers, which will be very useful for the development of action plan of collective action of disaster risk reduction intervention among transboundary stakeholders. Detail of study is deribed in the methodology section below.


## 2.3 Research Methodology

This study emphasized on recognizing transboundary flood risk drivers and classifying those drivers to gain insight into the priorities of drivers for structuring the priority of disaster risk reduction intervention. To exemplify this theoretical consideration the study used the Ciliwung River basin as the study location.

To have more realistic transboundary flood drivers / challenges, preliminary identification of all transboundary challenges was obtained through literature and documents review which was combined by field observation, from the perspective of upstream, midstream, and downstream flood issues and challenges. These challenges are also called as flood drivers in this study, which are used interchangeably according to the context of discussion. Results of preliminary identification is structured in a Source-Pathway-Receptor framework shown in Table 1 as well as in fish bone diagram shown in **Figure 1**.

To have a more realistic flood risk driver's framework, a Focus Group Discussion (FGD) among transboundary stakeholders/government institutions involved in Ciliwung river management was conducted.    With about 13

actors/stakeholders from national and transboundary cities and regencies of the Ciliwung River basin have actively participated in the FGD, the name of stakeholders can be seen also **Table 2**.

**Table 2 Selected Experts and Their Roles**

| No | Institution | Roles |
|---|---|---|
| 1 | National Planning and Development Agency (BAPPENAS) | Coordinates Development Planning and Financing for Transboundary Regions |
| 2 | Ciliwung-Cisadane River Basin Authority (BBWS Ciliwung Cisadane) | Executes Ciliwung River Program, i.e., flood control development and maintenance |
| 3 | Ministry of Spatial Planning (Kementerian ATR) | Coordinates Spatial Planning and Controlling for Transboundary Regions |
| 4 | Jakarta Provincial Planning and Development Agency (Bappeda Provinsi DKI Jakarta) | Coordinates Development Planning and Financing in Jakarta Provincial |
| 5 | Jakarta Provincial Water Resource Agency (Dinas Sumber Daya Air Provinsi DKI Jakarta) | Executes flood control development and maintenance in Jakarta Provincial |
| 6 | West Java Provincial Planning and Development Agency (Bappeda Provinsi Jawa Barat) | Coordinates Development Planning and Financing in West Java Provincial |
| 7 | West Java Provincial Water Resource Agency (Dinas Sumber Daya Air Provinsi Jawa Barat) | Executes flood control development and maintenance in West Java Provincial |
| 8 | Depok City Planning and Development Agency (Bappeda Kota Depok) | Coordinates Development Planning and Financing in Depok City |
| 9 | Depok City Public Work and Spatial Planning Agency (Dinas PUPR Kota Depok) | Executes flood control development and spatial planning in Depok City |
| 10 | Bogor City Planning and Development Agency (Bappeda Kota Bogor) | Coordinates of Development Planning and Financing in Bogor City |
| 11 | Bogor City Public Work and Spatial Planning Agency (Dinas PUPR Kota Bogor) | Executes of flood control development and spatial planning in Bogor City |
| 12 | Bogor Regency Planning and Development Agency (Bappeda Kabupaten Bogor) | Coordinates of Development Planning and Financing in Bogor Regency |
| 13 | Bogor Regency Public Work and Spatial Planning Agency (Dinas PUPR Kabupaten Bogor) | Executes of flood control development and spatial planning in Bogor Regency |

Result of FGD shows the final improved framework with 13 important flood drivers, as shown in **Table 3** below. This framework is then used as the basis for in-depth interview using several key questions:

- What are the significant issues related with each driver?
- How is the relationship of influence and dependence criteria between drivers from the perspective of flood disaster risk reduction intervention?
- Who are involved in implementing the flood disaster risk reduction intervention for each driver?
- What are the potential effect with dependence criteria of each drivers?

**Table 3 Identified Ciliwung Flood Risk Drivers**

| Codes | Key Flood Drivers | Modified Terms and Classification from Table 1 |
|---|---|---|
| A1 | Extreme Rainfall | Precipitation (Source) |
| A2 | Waste and Sedimentation | Sediment Supply (Pathway) |
| A3 | Drainage Capacity | Sewer Conveyance (Pathway) |
| A4 | River Capacity | River Morphology (Pathway) |
| A5 | Urbanization | Urbanization (Pathway) |
| A6 | Growth Population | Urbanization (Pathway)<br>Social Impact (Receptor) |
| A7 | Catchment Area | River Vegetation (Pathway) |
| A8 | Built Environment | Urban Impact (Receptor),<br>Rural Land Management (Pathway) |
| A9 | Ground Water Exploitation | Groundwater flooding (Pathway) |
| A10 | Stakeholder Cooperation and Coordination (Government, Lifelines, Business, Community) | Stakeholder Behaviour (Pathway) |
| A11 | Land Subsidence | Groundwater flooding (Pathway) |
| A12 | Spatial Plan | Environmental Regulation (Pathway) |
| A13 | Flood Controls / Structural Mitigation (Dams, Levees, Reservoirs, Water Pump, Dikes) | River Conveyance (Pathway) |

Through in-depth interviews to targeted stakeholders from national to local government as also listed in Table 2, the experts' judgements are used to establish a conceptual link among the drivers, where these drivers could also be impactful to each other. The drivers' relative responses were obtained by calculating the collected opinion in the interviews. Expert judgement assists in depicting the suitable interaction between these drivers. These variables' conceptual link is characterized using a pair-wise relationship as either "influencing" other drivers or being "influenced" by other drivers. Using VAXO symbols, four symbols which have been defined to demonstrate the linkage between i and j drivers, the expert judgment matrix called Structural Self Interaction Matrix (SSIM) were mapped to associate with 13 drivers. Table 2 shows the VAXO symbols. Symbol V represents driver i influencing driver j, symbol A shows driver j influencing driver i, symbol X shows driver i and j influencing each other, and symbol O shows driver i and j are not associated.

Beside Table 2 also shows the VAXO rule for converting SSIM values into IRM values to structure IRM Initial Reachability Matrix shown in Table 7, as the data set input for MICMAC model analysis.

The output from the MICMAC analysis reveals the classification of the drivers into four quadrants, i.e. independent, linkage, autonomous and dependents variables, see also Figure 4. This is then followed with discussion using more information from the stakeholders during interviews.

**SSIM and IRM Matrix for MICMAC Analysis**


To recognize the challenges and complexity of the transboundary management in flood risk reduction, this study involves a qualitative and quantitative approach using primary data from stakeholder focus group discussion, in-depth interviews, and field observations. The 2019 Focus Group Discussion on "Mitigating Hydrometeorological Hazard Impacts Through Transboundary River Management in the Ciliwung River Basin" aims for sensitisation of stakeholders, which has helped to build trust and forge a potential pathway to impact through river basin management policy. The interviews and field

observation were conducted from September until December 2020; while amidst that period, there were several Ciliwung flood events that occurred in Greater Jakarta, making the obtained data more relevant and up to date. Face-to-face and online interviews are used for the interview methods as the consequence of Indonesia's Large Scale Social Restriction due to the Covid-19 Pandemic. Both kinds of interview methods perform the same quality of content.

To have reliable results in a case of transboundary river management, the interviews included multi-level governments, i.e., national, provincial, and city/regency governments, along Ciliwung River Basin. The role of river basin-related institutions could be divided as follows (Dewi & Ast, 2017): first is the institution with main role in the planning process at each of the national, provincial, and city/regency levels; second is the institutions responsible for the implementation process of flood management projects, also at each of the national, provincial, and city/regency levels; institutions that have the power of

coordination.

Based on these two criteria, thirteen experts related to Ciliwung river flood from different levels and regions are selected as the target respondents, see also **Table 2.**

Given a set of flood risk drivers as presented in **Table 1** all experts, as listed in **Table 2**, were asked to identify the key flood risk drivers of the Ciliwung River flood and its interrelations with justifications. They were also asked to explain the actual

condition of each driver based on their empirical knowledge and scope of work. Grounded theory, as the qualitative method, is then used to interpret experts' statements into codes. The grounded theory method involves gathering and analysing data to generate a middle-range theory (Charmaz, 1995). Analytic processes consist of data coding, developing, checking, and integrating theoretical categories, and constructing analytic narratives (Glaser & Strauss, 2017).

Through in-depth interviews to targeted stakeholders from national to local government as shown in **Table 2**, the experts'

judgements are used to establish a conceptual link among the drivers, where these drivers could also be impactful to each other. The drivers' relative responses were obtained by calculating the collected opinion in the interviews. Expert judgement assists in depicting the suitable interaction between these drivers. These variables' conceptual link is characterized using a pair-wise relationship as either "influencing" other drivers or being "influenced" by other drivers. Using VAXO symbols, four symbols which have been defined to demonstrate the linkage between i and j drivers, the expert judgment matrix called

Structural Self Interaction Matrix (SSIM) were mapped to associate with 13 drivers. Error! Reference source not found. shows the VAXO symbols. Symbol V represents driver i influencing driver j, symbol A shows driver j influencing driver i, symbol X shows driver i and j influencing each other, and symbol O shows driver i and j are not associated.

Beside Error! Reference source not found. also shows the VAXO rule for converting SSIM values into IRM values to structure IRM Initial Reachability Matrix shown in Error! Reference source not found., as the data set input for MICMAC model

analysis. Then output of MICMAC analysis showing the classification of the drivers into four quadrants, i.e. independent, linkage, autonomous and dependents variables, see also **Figure 4**. This is then followed with discussion using more information from the stakeholders during interviews.

**Table 4 SSIM and IRM Value**

| Relation | SSIM Symbols | IRM value |
| --- | --- | --- |
| **driver i influencing driver j** | V | 1 |
| **driver j influencing driver i** | A | 1 |
| **drivers i and j influencing each other** | X | 1 |
| **drivers i and j are not associated** | O | 0 |


**Table 5 SSIM Matrix of Experts' Judgements**

| i/j | A1 | A2 | A3 | A4 | A5 | A6 | A7 | A8 | A9 | A10 | A11 | A12 | A13 |
|-----|----|----|----|----|----|----|----|----|----|-----|-----|-----|-----|
| A1 | V | 0 | V | V | 0 | 0 | 0 | 0 | 0 | 0 | 0 | 0 | V |
| A2 |   | V | V | V | 0 | 0 | 0 | A | 0 | A | 0 | 0 | V |
| A3 |   |   | V | V | 0 | 0 | A | 0 | 0 | 0 | 0 | A | V |
| A4 |   |   |   | V | 0 | 0 | A | 0 | 0 | A | 0 | A | X |
| A5 |   |   |   |   | V | V | 0 | 0 | 0 | 0 | 0 | 0 | 0 |
| A6 |   |   |   |   |   | V | 0 | V | 0 | 0 | 0 | V | 0 |
| A7 |   |   |   |   |   |   | V | A | 0 | A | 0 | A | A |
| A8 |   |   |   |   |   |   |   | V | V | 0 | 0 | A | 0 |
| A9 |   |   |   |   |   |   |   |   | V | 0 | V | 0 | 0 |
| A10 |   |   |   |   |   |   |   |   |   | V | 0 | X | V |
| A11 |   |   |   |   |   |   |   |   |   |   | V | 0 | V |
| A12 |   |   |   |   |   |   |   |   |   |   |   | V | V |
| A13 |   |   |   |   |   |   |   |   |   |   |   |   | V |

**Table 6 IRM Matrix**

| i/j | A1 | A2 | A3 | A4 | A5 | A6 | A7 | A8 | A9 | A10 | A11 | A12 | A13 |
|-----|----|----|----|----|----|----|----|----|----|-----|-----|-----|-----|
| A1 | 1 | 0 | 1 | 1 | 0 | 0 | 0 | 0 | 0 | 0 | 0 | 0 | 1 |
| A2 | 0 | 1 | 1 | 1 | 0 | 0 | 0 | 0 | 0 | 0 | 0 | 0 | 1 |
| A3 | 0 | 0 | 1 | 1 | 0 | 0 | 0 | 0 | 0 | 0 | 0 | 0 | 1 |
| A4 | 0 | 0 | 0 | 1 | 0 | 0 | 0 | 0 | 0 | 0 | 0 | 0 | 1 |
| A5 | 0 | 0 | 0 | 0 | 1 | 1 | 0 | 0 | 0 | 0 | 0 | 0 | 0 |
| A6 | 0 | 0 | 0 | 0 | 0 | 1 | 0 | 1 | 0 | 0 | 0 | 1 | 0 |
| A7 | 0 | 0 | 1 | 1 | 0 | 0 | 1 | 0 | 0 | 0 | 0 | 0 | 0 |
| A8 | 0 | 1 | 0 | 0 | 0 | 0 | 1 | 1 | 1 | 0 | 0 | 0 | 0 |
| A9 | 0 | 0 | 0 | 0 | 0 | 0 | 0 | 0 | 1 | 0 | 1 | 0 | 0 |
| A10 | 0 | 1 | 0 | 1 | 0 | 0 | 1 | 0 | 0 | 1 | 0 | 1 | 1 |
| A11 | 0 | 0 | 0 | 0 | 0 | 0 | 0 | 0 | 0 | 0 | 1 | 0 | 1 |
| A12 | 0 | 0 | 1 | 1 | 0 | 0 | 1 | 1 | 0 | 1 | 0 | 1 | 1 |
| A13 | 0 | 0 | 0 | 1 | 0 | 0 | 1 | 0 | 0 | 0 | 0 | 0 | 1 |

## 3. Example of the Ciliwung River Basin, Jakarta Indonesia

### 3.1 Ciliwung River Basin and Jakarta Metropolitan Development

Jakarta Metropolitan Area, known as Greater Jakarta, is an agglomeration city of Jakarta-Bogor-Depok-Tangerang-Bekasi. This metropolitan has been one of the most appealing locations for both domestic and foreign investment, with a large number of entrepreneurs and skilled laborers, as well as high access to decision-makers (Firman, 1998). To compare, as of 2023 Tokyo Metropolitan Area in Japan was the largest world urban agglomeration, with 36.57 million people living in 7,693 km$^2$, while Jakarta Metropolitan Area ranked the second with 34 million people living in 7,315 km$^2$ area (Dyvik, 2023; Rahayu, 2022a). According to Euromonitor International, Jakarta Metropolitan will become the most prominent mega city globally, with

estimated population of 35.6 million by 2030 (Dyvik, 2023). As defined by the UN, megacity is a city with more than 10 million population.

Jakarta Metropolitan is geographically crossed by 13 river systems, including Ciliwung river which has the longest and biggest river basin. According to Presidential Decree Number 12/2012, the determination of Ciliwung River Basin Area is 140 kilometres long with nearly 438 km$^2$ catchment area crossing Jakarta Province and three major cities and one regency in West Java Province. As shown in Figure 2a and 2 b, the land use for Ciliwung river basin can be divided into three zone, i.e. upstream, midstream and downstream. Bogor Regency and Bogor City are in the upstream aiming for conservation area and water resources, while Depok City is in the middle stream aiming for buffer zone and catchment area, and Jakarta Province is in the downstream aiming for cultivation and coastal protection.

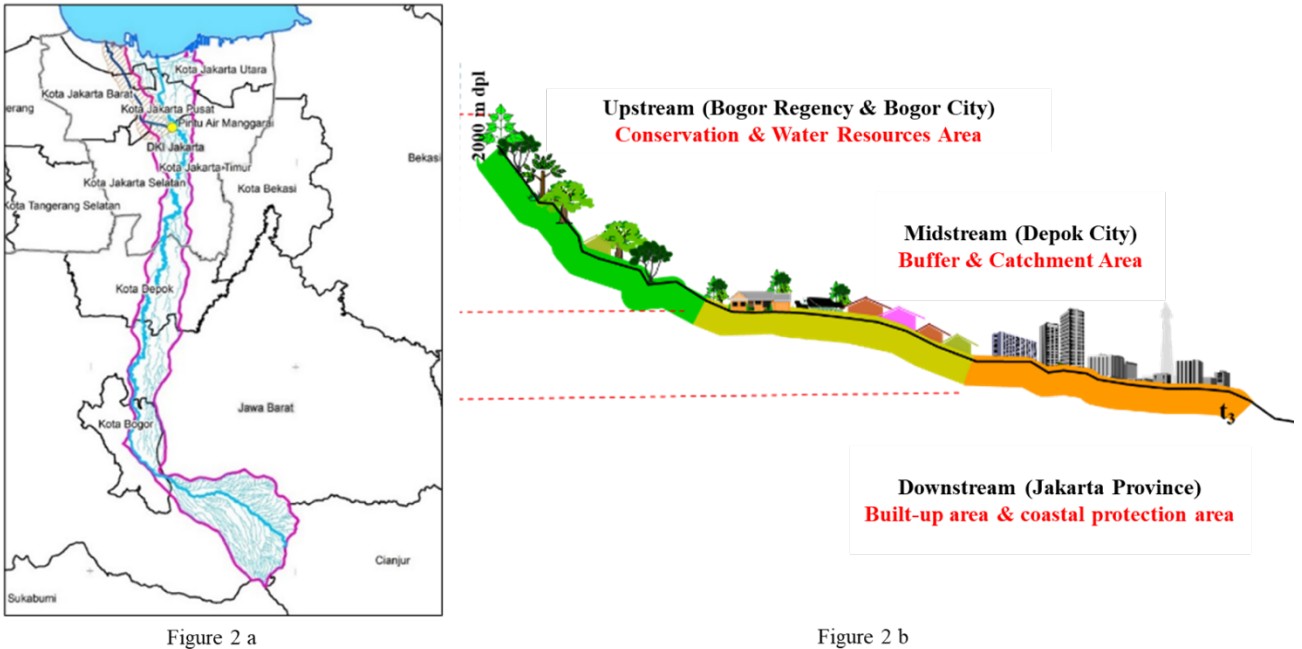

Figure 2 a           Figure 2 b

**Figure 2 Land Use in Ciliwung River Basin (source: FGD, 2020 in** Rahayu, 2022b**)**

Being situated in a watershed, Greater Jakarta area is exposed to hydro-meteorological hazards, such as flood. Severe floods due to the Ciliwung River's overflow have been recorded in 1996, 2002, 2007, 2012, and 2013 (Dewi & Ast, 2017), while the most severe took place in early 2020. A total of 13 administrative units in Jakarta, Banten Provinces, and West Java Provinces were flooded, causing an estimation of loss around IDR 5.2 trillion during that 2020 event. The coastal region of Greater Jakarta is also exposed to frequent tidal flood.

From the flood vulnerability perspective, approximately 25 million people lives in the Ciliwung River Basin (CRB) with its growth rate around 1.4 percent. The flood-prone areas are primarily located in the densely populated Jakarta Province (DKI Jakarta), with about 28,818 households by 2009 (Rahayu & Nasu, 2010), and about 34,051 households by 2021 (DKI Housing and Building Department).

Several factors have been attributed to the increased magnitude of flood impacts in Greater Jakarta over the past few decades, including precipitations, land-use change, sea level rise and land subsidence (Budiyono et al., 2016). However, massive urbanization in Jakarta between 1995-2014 has significantly decreased runoff regulation, green spaces and bodies of water, as well as affected landscape pattern changes (Maheng et al., 2021), and has strongly influenced spatial characteristics, such as industrial parks, mixed-use new towns and large-scale residential areas, and shopping centres (Firman & Fahmi, 2017). As identified by a previous study (Silver, 2007), the Jakarta land use change was initiated in the early 1980s, where many agricultural and forest area in suburban Jakarta were transformed into large-scale subdivisions and new towns.

More than 30 large new suburban towns and industrial parks were built in the peripheries of Jakarta City between 1990 and 2010, with average size from 500 to 30,000 hectares (Firman, 2014). As a consequence of these vast peri-urban development,

massive conversion of water catchment area, wetland, and green areas have been occurred, which has also led to an increased flood threat to Jakarta. Further, the urbanization was clearly seen to increase not only the intensity and volume of inundations, but also the runoff, river flow discharges, which all lead to the increase of flood threat (Priyambodoho et al., 2022). Thus, identifying flood risk drivers must address spatial scales from the upstream to the downstream (Dawson et al., 2009). Figure 3 below shows the rapid land use change in three decades, from 2000 to 2017.


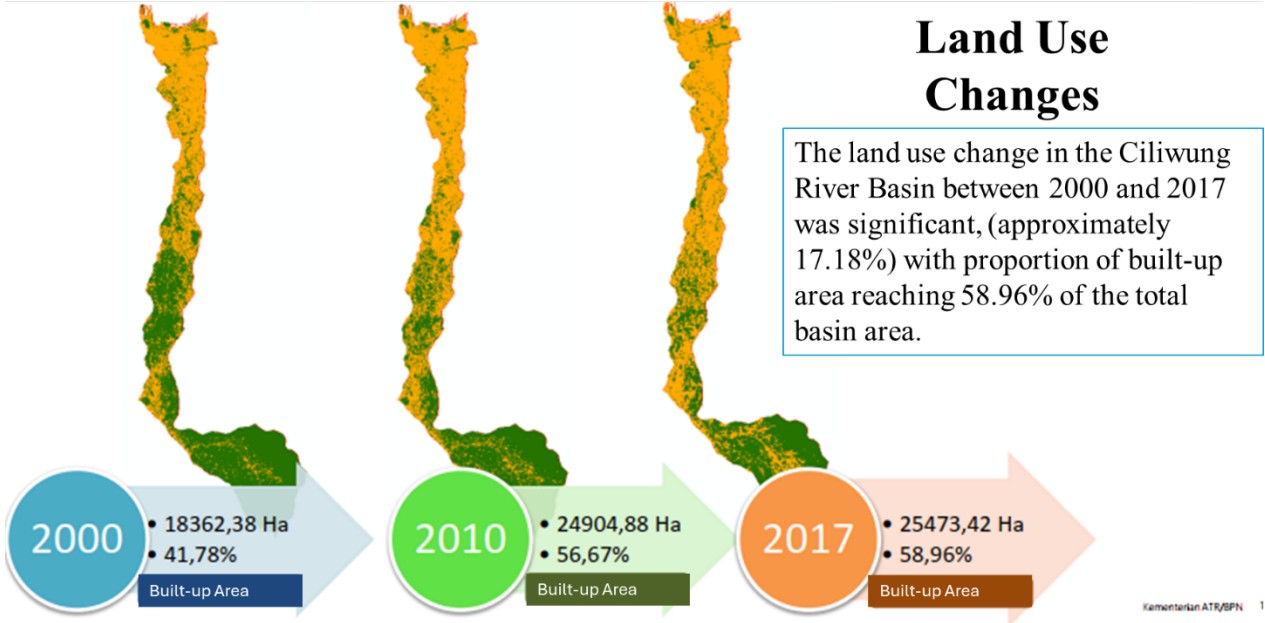

**Figure 3 Ciliwung River Basin Land Use Change (Source; ATR, 2020 in** Rahayu, 2022a**)**

In the last two decades, several scholars have studied flood drivers in the Ciliwung River Basin. For instance, Texier (2008)

identified the root causes of disaster vulnerability in Jakarta Province, Emam et al. (2016) studied the effect of climate and land use change in the Upper Ciliwung River, meanwhile both Asdak et al. (2018) and Texier (2008) have analysed problems in downstream flooding, and Sagala et al. (2013) highlighted Greater Jakarta flood vulnerability. Of those studies, however, no one discussed the issues of transboundary flood risk drivers, responsibility sharing, as well as mentioned which drivers are the most critical for the transboundary flood risk management.

The Ciliwung River Basin Authority (BBWS-CC), under Ministry of Public Works, bears responsibility of the whole Ciliwung River Basin management. The complexity of transboundary flood risk drivers has come in those several regions of the river basin with many aspects consequently as part of the megapolitan development. Flood risk managed by a single national authority is indeed complicated enough and becomes much more complex when dealing with the transboundary river. To protect the people at risk toward disaster is under responsibility of local government (Law No. 24 / 2007 Regarding

Disaster Management, 2007; Law No. 23 / 2014 Regarding Local Government, 2014).

**3.2 Transboundary flood risk driver framework for CRB**

As the downstream area, Jakarta City experienced more severe floods compared to other regions due to its geographical condition. Despite extensive efforts by the Dutch and Indonesian governments, Jakarta is still prone to flooding since its

location in a major river delta (Asdak et al., 2018). Due to land subsidence issues, about 40% of Jakarta City areas are a few meters below sea level, estimated to be 1 to 15 centimetres rates per year both spatially and temporally (Latief et al., 2018).

The existing Jakarta flood control system was developed based on Prof. H. Van Breen's (1973) concept, in which the overflow rainwater from outside Jakarta redirected via flood canals (West Flood canals and East Flood Canals) that circle Jakarta. Run-off within the city of Jakarta is discharged through local drainage system by gravity and discarded with a polder system, including water pump and pond retention in low areas (Kusuma et al., 2010).

Apart from natural causing factors, rapid urbanization and massive growth in population led to an increase in the susceptibility and vulnerability to Jakarta floods (Rahayu and Nasu, 2010). The rapid growth of urban sprawl (Maheng et al, 2021, Firman and Fahmi, 2017) has caused massive land conversion from the catchment area to the built environment. A change in land use over time can have significant effects on run-off (Mishra et al., 2018). Uncontrolled land-use change due to poor spatial planning along the Ciliwung River Basin makes the flooding becoming more complicated to handle (Asdak et al., 2018). To control current developments and minimize future risks, strong governance with good long-term spatial planning is needed (Rahayu et al., 2019). It is expected that spatial planning will contribute to flood mitigation in floodplain areas (White & Howe, 2002; White & Richards, 2007) by regulating the land use types, spatial pattern, development scales, and physical structure designs. It can affect the likelihood of floods and its consequential damage (Neuvel & van den Brink, 2009; White & Richards, 2007).

The emergence of Law 23/2014 regarding Local Government resulted in the right and obligatory sharing between national and local government, known also as decentralization. Decentralization and power-sharing expanded disaster management responsibility at local levels with national policy impacting it (Sunarharum et al., 2021). Since Ciliwung River Basin flows along transboundary regions, therefore national, provincial as well as city/regency governments along Ciliwung River Basin are responsible for flood risk management as well. While governments may be able to mitigate flood risk, communities, especially those affected by floods, must be included in flood risk management decisions (Faulkner et al., 2007). However, as flood risk management involves various stakeholders (i.e., governments, communities, academics, media, and privates) and multiple objectives, conflicts may also arise. Up until recently, the coordination among stakeholders in the Ciliwung River Basin still meets many challenges and as a result, affects the decision-making (Sunarharum et al., 2021).

The Ministry of Public Works Regulation Number 13/PRT/M/2006 regarding River Management has defined that the Ciliwung River is a transboundary river that crosses two provinces and four cities/regencies, and is controlled by the National Government, i.e., Ciliwung-Cisadane River Basin Authority (BBWS CC), in collaboration with local stakeholders. Understanding the transboundary management in flood risk reduction is very critical.

Based on input from the Stakeholders Focus Group Discussion (FGD) conducted by this study on 26 September 2019 under collaboration with National Planning Agency (Bappenas), the preliminary framewok of Ciliwung River flood drivers shown in **Table 1** and combined with **Figure 1** were refined with few modifications in terms from the dicussion and recommendation of FGD. The FGD was attended by 12 institutions directly and indirectly involved in Ciliwung Flood Management, from national, provincial, and local government. The final result of improved Ciliwung flood risk driver's framework is shown in **Table 3** with thirteen independent flood driver variables.

## 4. Result and Discussion

The result of MICMAC model is shown in **Figure 4** diagram below. The 13 flood drivers are mapped into four quadrants, i.e., independent, linkage, autonomous, and dependence. Through this analysis, two flood drivers, i.e., Spatial Plan (A12) and Stakeholders Cooperation and Coordination (A10), are emerged as critical flood risk and the most powerful drivers in Ciliwung River Basin. These two drivers are independent drivers, they have the highest driving power and the lowest dependency power; they exert a significant influence on other driver from other groups. These two drivers will take important role in overall system of transboundary Ciliwung river flood management. The intervention on these two factors will have the highest impact to other flood drivers. However, this study found no variables fall in linkage groups. This means that all the stated variables are stable.

Meanwhile seven flood drivers are autonomous drivers. They are Extreme Rainfall (A1), Built Environment (A8), Growth Population (A6), Waste and Sedimentation (A2), Urbanization (A5), Ground Water Exploitation (A9), and Land Subsidence (A11). These seven autonomous drivers are characterised by extremely low driving power and low dependency to other drivers' group, they are more self-contained. However, the Extreme Rainfall (A1) and Built Environment (A8) are located closer to independent group, having relatively high driving power. These two will be also discussed further in the next section. The rest of flood drivers show the least driving ability and the high reliance on other groups, they fall in the dependence group. They are drainage capacity (A3), catchment area (A7), flood control-structural mitigation (A13), river capacity (A4). These factors are the weakest in the key flood driver mapping, relying heavily on other drivers. Any action taken by other dirvers will have an impact on these dependence drivers.. River capacity appears as the most dependent drivers, indicating the end of the flood driver chain in the overall system of transboundary Ciliwung river flood management.

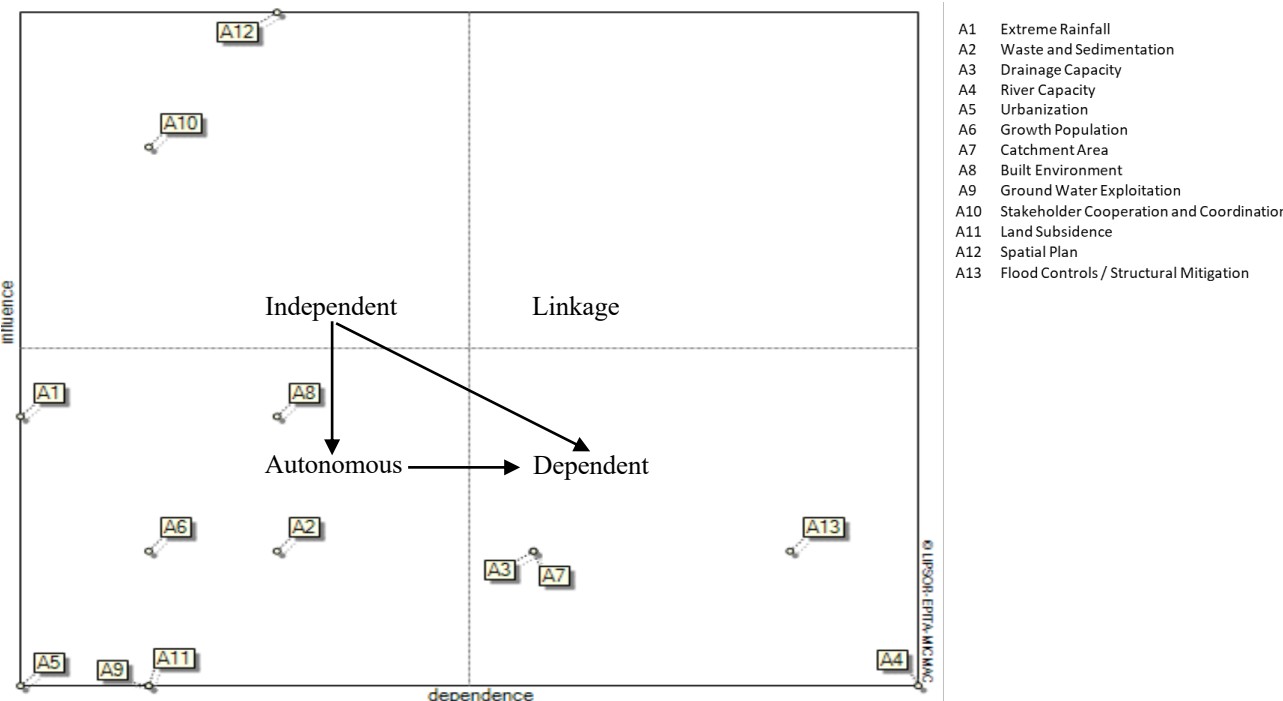

**Figure 4 MICMAC model result diagram**

Through MICMAC analysis, this study has unveiled the important of two key flood drivers, i.e., non- structural mitigation through spatial planning and stakeholders' cooperation and coordination, covering the issues and challenges of Ciliwung flood risk governance in reducing/managing the Ciliwung River flood risk and achieving sustainable catchment area. Referring to Source-Pathway-Receptor model in **Table 3**, it is found that in the case of Ciliwiung River flood risk management, both "pathway' such as stakeholder behaviour and environmental regulation give the main influence in the set of flood risk drivers. Surprisingly, this study also unveiled that the most dependence drivers are found related with the catchment area and capacity to channel out the flood water, such as drainage capacity, river capacity and flood control. Any action taken on other drivers will give significant impact on these dependence drivers.

Findings of this study could be as the inputs for other cities/metropolitan which has similar problem transboundary of river flood management. Further, the two highest key flood drivers, i.e., spatial planning and stakeholders' cooperation and coordination, the moderate high key flood driver, i.e., extreme rainfall, as well as the most dependence flood drivers, i.e., flood controls, will be discussed.

**4.1 Spatial Plan**

A holistic approach is needed in the development of flood risk management, covering the upstream, midstream, and downstream. Controlling the river basin development through integrated spatial planning from upstream, midstream, and downstream are necessary to reduce flood risk in the river basin. Spatial planning is a system concerned with long- or medium-term objectives and strategies for the regions development, and consists of planning processes, space utilization and control of space utilization. The spatial plan basically is composed of spatial structure and spatial pattern. The spatial structure

arranges residential and business centres, and all supporting infrastructure networks and facilities for related socio-economic activities of the community. The spatial pattern distributes space allocation by considering protection functions as well as cultivation and/or development function.

Since Ciliwung river is one of the longest river basins passing through 2 province and 5 regencies/cities, the role of transboundary coordination in water resource and flood risk management become very critical from the perspective of ego

sectoral as well as ego area jurisdiction.

This is in line with the findings of the study, that shows the spatial plan (A12) and stakeholder cooperation and coordination (A10) are the most critical flood risk driver in Ciliwung River Basin management. Both key flood drivers are independent drivers having the most powerful influence on other key flood drivers.

Most flood risk reduction regulations are included in each city/regency as well as provincial spatial plans, such as land use

regulation, structural mitigation development, catchment area preservation, and river maintenance. The integration of these disaster risk reduction countermeasures into spatial planning regulation have studied by several works. For example, flood risk avoidance is often used for controlling spatial development in floodplain area including relocation plan (Kang et al., 2009) while flood risk defence is for preventing the region from flood water by building river dykes (Voorendt, 2017) Meanwhile flood risk mitigation is for reducing flood impact loss by structural mitigation or nature base solution for flood

detention (Sayer et al, 2013), flood retention (Wingfield et al., 2019), and flood passages (Kang et al., 2009). Last but not least, the flood risk preparedness is used for evacuation plans and flood risk recovery is used for developing post recovery plan and critical infrastructure protection (Meng et al., 2022; Sayers et al., 2013).

In the case of Ciliwung river, these city/regency as well as provincial spatial plan often developed only to suit their own needs and objectives, without considering broader needs, such as for regional flood risk management. To bridge the transboundary

issues and challenges of spatial planning as tools for flood risk management, the Ministry of Spatial Plan established the Greater Jakarta Spatial Plan in Government Law Number 60/2020 to incorporate all those related local and provincial spatial plans. This Greater Jakarta Spatial Plan is expected to function as transboundary spatial plan. However, until now, it is not fully enforced yet for water resource and flood risk river management.

Further, Ciliwung-Cisadane River Basin Agency has formulated a plan in the Ministry of Public Works Decree No

26/KPTS/M/2015 regarding integration of Ciliwung and Cisadane river basins, which means also combining both Ciliwung-Cisadane river management programs. During the FGD, the Ciliwung-Cisadane River Basin Authority (BBWS Ciliwung Cisadane) stated that "There were massive land use change at the upstream of river basin, where this became main flood driver. However, the flood itself was worsened by the extreme rainfall". Ministry of Environment and Forestry (KLHK) – Agency for Citarum-Ciliwung River Basin Management (BPDAS – Balai Pengelolaan Daerah Aliran Sungai Citarum-Ciliwung)

assumed that the land use and land cover change become primary contributor to Ciliwung River Basin flood. For example, the deforestation of the upstream Ciliwung River Basin may cause flooding on the midstream and the downstream. Thus, the spatial plan needs to consider the land use change as well as the flood risk reduction.

National Planning and Development Agency (BAPPENAS) said that starting from 2010, average runoff coefficient has been recalculated due to land use change. There was an increase up to 0.4 and 0.5 in 2014. Meanwhile, West Java Planning and

Development Agency (Bappeda Jabar) assumed that the development in the catchment area should consider transboundary

commitment for rehabilitation of conservation and protected area. The more land use change, the less water catchment area, which lead to flood.

During FGD, several experts judge that development control in the Ciliwung River Basin appeared to be weak as land-use changes emerged at the upstream, midstream, and downstream. Urban and regional development is increasingly not considering catchment area provision over time due to economic pressure. Although spatial plans have been created on many levels, development control remains powerless to retain the catchment area. Some upstream regions have been turned into residences, hotels, villas, and restaurants, resulting in increased water run-off, but in Jakarta, new settlements are created without regard for spatial planning, affecting water absorption.

Since, it appears from the results of the MICMAC model that spatial planning has significantly affected drainage capacity, river capacity, catchment area, built environment, stakeholder cooperation, and flood control development, thus improving, strengthening and integrating the spatial plan related with Ciliwung river basin of Jakarta Province, Depok City, Bogor City and Bogor Regency is main priority in order to have better result of overall sustainable flood risk management.

Land use plans are supposed to substantially impact the basin's development (S. Y. Wang et al., 2010). Therefore, spatial planning is a critical tool for reducing flood risk (Neuvel & van den Brink, 2009). Budiyono et al. (2016) investigated the great potential for urban planning to mitigate flood risk. It demonstrates that if Jakarta's land use follows the 2030 spatial plan, flood risk will be reduced by 12%. This highlights the great potential of land use planning for flood risk reduction.

Having a solid and integrated spatial plan is not enough to reduce flood risk unless followed by robust development control. Strict development control must be applied at the basin level, which means not only in Jakarta Provincial but also in Depok City, Bogor City, and Bogor Regency as part of the Ciliwung River Basin regions.

Development control regulation in the Ciliwung River basin may differ from upstream to downstream municipality depending on physical, environmental, and institutional characteristics. Development control in the upstream area mainly aims to preserve the catchment area, while in the downstream area, it primarily aims to prevent groundwater exploitation and higher physical vulnerability. The national government, along with the local government, must create tight instruments for development control, while the local government itself must carry out strict surveillance and give penalties for all development violations.

To strengthen development control in the Ciliwung River basin, a holistic regulation regarding development control (mechanism, instrument, zoning technique, and executor (i.e., task force)) in the Ciliwung River basin level must be legalized as a national policy. The President should make this a national priority program, considering the areas impacted and the number of losses that the Ciliwung River flood has generated. Up until recently, integrated development control policy in the Ciliwung river basin has not been developed yet, even though there has been the Greater Jakarta Spatial Plan (Law of President Number 60/2020) is expected to be holistic transboundary spatial plan and Ciliwung-Cisadane River Basin management program (Ministry of Public Works Decree No 26/KPTS/M/2015).

However, difficulties may arise in their management and governance when dealing with transboundary river management. Several critical institutions are involved in flood risk reduction, i.e., Governments at national, sub-national, and local levels, utility companies, private businesses, and community groups (Jha et al., 2012). Coordination is required both between actors at different authority levels (vertical coordination) as well as among actors within administrative boundaries (horizontal coordination).

Meanwhile, as transboundary river basin governments have many flood drivers to overcome, it has to meet concrete criteria to have an appropriate arrangement for river basin management. Firstly, clear roles and responsibility-sharing among river management institutions are essential for effective coordination (Jha et al., 2012). A negotiation procedure and coordination mechanism are required (Barbazza & Tello, 2014). Secondly, a coordination mechanism is also important to enhance information and data flows and coordinate decision-making and implementation. Thirdly, leadership and power to enforce coordination contribute to the fragmentation of the institutional arrangement (Brown, 2005). Therefore, an effort to make a governance forum in the Ciliwung River Basin must follow those three criteria to waive ineffectiveness.

According to Millington (2006), there are two types of river basin forum. Those are (1) the river basin coordinating committee and (2) the river basin commission. A river basin coordinating committee is formed for stable and mature river basin management. This model mainly relies on the fair cooperation and participation of its members. The committee has no executive authority and cannot override the member organization's tasks and operations. The coordinating committee would be made up of major water-related agencies from each of the basin's states.

Further, when problems happen frequently, a river basin committee is formed. A basin commission is a more formalized group than a committee. It would consist of a management board that would establish objectives, goals, policies, and strategic direction. The commission would be supported by a technical office of water, natural resource, socioeconomic planning, and management experts, many of whom would be drawn from existing agencies in the basin. To offer ultimate power, a Ministerial Council could lead the commission, and the basin commission would then focus on strategic natural resource management of the rivers and catchments. The fact that Ciliwung River Basin management still meets conflict and is not stable in management attests that the river basin commission might be the fittest model for Ciliwung river basin governance.

## 4.2 Stakeholder Coordination and Cooperation

As the second critical key flood drivers, the stakeholder cooperation and coordination (A10) are playing important role in transboundary Ciliwung River Basin management. It is independent drivers which has the most powerful influence on other key flood drivers. A river basin sustainable development and management need to consider the basin as a whole, with multiple interactions of water-ecosystem-economy from the upstream, midstream, and downstream areas (Cheng et al., 2014). Meanwhile, Lorenz et al (2001) defined sustainable river basin management through interaction model among social capital, human capital, natural capital and man-made capital. This model generated laws, regulations, information flow for triple helix stakeholders, i.e., government, business and community.

Based on the interviews in this study, the stakeholder cooperation and coordination among governments, communities, academics, lifelines, and business significantly affect other drivers, such as waste and sedimentation, river capacity, catchment area, built environment, groundwater exploitation, spatial and development plan, and flood controls.

The imperative issue takes place in coordination and cooperation among governments. The transboundary governance forum for Ciliwung River Basin management has been reformed many times. Two previous forms are the Ciliwung River Basin Forum in 2007, led alternately by the Governor in the Ciliwung River Basin, and the Ciliwung Water Resource Management Coordination Team in 2011, led alternately by each of the Planning and Development Agency in the Ciliwung River Basin. ccording to experts' experience obtained during FGD, the reformation keeps occurring due to the ineffectiveness of the forum mechanism, the powerless leader, and the conflict of interest. There was no clear framework for the forum and no legal agreement about how coordination and cooperation among institutions should work. This eventually resulted in no clear action. Each institution merely understands its own jobs and pays attention to their interest or said as sectoral egos. Also, there was no strong figure who could lead the forum.

Meanwhile, the Ministry of Environment and Forestry (KLHK) – Agency for Citarum-Ciliwung River Basin Management (BPDAS – Balai Pengelolaan Daerah Aliran Sungai Citarum-Ciliwung) stated that Ciliwung river is very complex in its governance. Jakarta Provincial Government responsibility is at the downstream, while West Java Provincial Government responsibility is at the upstream only. The transboundary river management should have one leader. If the Ministry of Environment and Forestry (KLHK) take a lead, it will be difficult for managing the inter and cross sectoral issues. For example, good lesson learned from the management of Citarum River, which was assigned to Citarum Harum lead by National Military based on the presidential decree. Thus, the Ciliwung river management needs to have similar governance structure, since the existence of Coordinating Board for Jakarta Metropolitan Area Development (BKSP Badan Kerja Sama Pembangunan) has not been optimal and sustainable.

To compare with Brantas River Basin tranboundary governance in East Java Indonesia, crossing several area jurisdictions, there has been a good water governance of the basin development and management handled by the central government (Brantas River Basin Executing Agency-BRBEA, known as the Brantas Project-BP), collaborated with state own enterprises, i.e., PT Indra Karya and PT Brantas Abipraya, since its beginning of 1960s with massive constructing projects of 8 large dams, 6 barrages, and rubber dams along the Brantas river for Irrigation, hydropower, flood control, and recreation purposes (Roestamy

& Fulazzaky, 2022). However, its main challenges are related with technical issues, institutional frameworks, and regulatory instruments to fulfil the needs of various stakeholders from various area jurisdiction. Water Resource Law 7/2004 has formalized the paradigm shift from project oriented to integrated river basin development, beside created, and empowered the institutional framework. Therefore by 2007, BRBEA has been fully taken over by national institutions in handling strategic issues of Brantas River Basin.

Meanwhile, National Planning and Development Agency (BAPPENAS) said that the transboundary coordination function for Ciliwung River could be managed by the Project Management Officer (PMO) of Jakarta Metropolitan Area (Greater Jakarta), which will be later substitute the Coordinating Board for Jakarta Metropolitan Area Development (BKSP) in coordinating the three provinces, i.e., Jakarta, West Java and Banten provinces. However, this seemed to be in effective since there is no involvement of national government. In the future, it is expected that there is coordinating body lead by Minister of

Agrarian Affair and Spatial Planning with Minister of National Planning and Development Agency as the vice, with its think tank at the existing Coordinating Board for Jakarta Metropolitan Area Development (BKSP).

### 5.3 Extreme Rainfall

Even though, the extreme rainfall (A1) is acknowledged in this study as an autonomous driver with moderate driving power and lowest dependency power. It is supposed to be characterised by self-contained and do not significantly drive or depend on

other drivers. However, the position the extreme rainfall in autonomous group is closer to independent group.   It means that this driver has relatively moderate influence power to affect other drivers. For example, the extreme rainfall may cause flood due to insufficient drainage capacity, poor catchment area, inappropriate flood controls/structural mitigation, and poor river capacity due to illegal settlement at riverbanks, trash sedimentation at the river and several other drivers.

In the last few decades, the climate change has impacted to the pattern of rainy season, as well as to the occurrence many

extreme precipitations which caused severe flood. The fact that climate change is likely a factor contributing to the heavy rainfall has been discussed by many scholars, i.e., intensified short duration heavy rainfall (Tamm et al., 2023). Compounded with vulnerability factors, such as socio-demographic, economic, physical and environment factors, this heavy rainfall will lead to the increase the flood risk.

Extreme precipitation is known to significantly affect Greater Jakarta floods (Mishra et al., 2018). Sudden changes of extreme

precipitation in short-duration precipitation, which lead the water volume, intensity, duration, and location may cause severe flood (O'Donnell & Thorne, 2020). Ciliwung River Flood occurred not only by upstream precipitation but also due to downstream rainfall. According to rainfall spatial distribution data, most of the Jakarta floods were caused by evenly distributed rainfall along the Ciliwung River Basin (Farid et al., 2021).

Floods occur when rivers capacity (A4) and drainage system (A3) do not have sufficient capacity to pass flow rates from

upstream to downstream (Asdak, 1995). The narrowing of the Ciliwung River's capacity is due to sedimentation and waste, as well as the construction of settlements on uncontrolled riverbanks. To decrease the flow rate, in a higher area, several infrastructures such as Situ (Lake) and Dams in Bogor Regency and Depok City, are built to control flood peak discharge in the up-stream and mid-stream areas of the Ciliwung River Basin (Nugraheni et al., 2020).

The impact and severity of these drivers are likely to increase, making changes in transboundary policy, planning, practice,

and coordination among the responsible agencies imperative across jurisdiction. Several works on climate change adaptation and flood disaster risk reduction in Jakarta conducted were identified (Rahayu et al., 2020).

## 4.4. Flood Controls / Structural Mitigation

The study found that flood control and structural mitigation is a driving factor with lowest driving power but highly dependency power. This means that this driver is the weakest in the key flood driver mapping, relying on the influence of other drivers. It gets easily inclined by other strong drivers. The experts' judgments were only focused on structural flood control, such as building dams, levees, dykes/flood canals, reservoirs, polders, and pump system.

Unless the flood control/structural mitigation are mainstreamed in spatial planning, the sustainable flood risk management will not be achieved (Meng et al., 2022). However, Jakarta has the highest people living at flood risk area, i.e., flood plain area, riverbanks. Alerting those people at risk before flood is significantly important. The existing flood early warning system needs to me improved by adopting people centre early warning system (Rahayu et al., 2020), by having reliable the four components, i.e., Monitoring and Warning Service, Dissemination & Communication, Risk Knowledge, and Response Capability. The first one is part of upstream component, while the rest are downstream component.

The existing flood monitoring and detecting in most river in Jakarta including Ciliwung River relied on manual water level measurement at the flood gate or dams. The real time advance monitoring, detecting and well as real time impact-based flood model are ongoing process of development for Greater Metropolitan Jakarta area. Alerting the people need to be advanced from the existing sirens, CCTV, community-based alerting system (Rahayu & Nasu, 2010). The community response and readiness of the stakeholder toward flood warning should be improved and tested regularly.

## 5. Conclusion

This study addresses broader issues of key flood driver in transboundary flood risk management. The Ciliwung river floods are very complex and influenced by tremendously development as a megapolitan region. This development has contributed to complex water-resource issues, such as increased flooding. Since the previous Greater Jakarta flood impacts trillion IDR losses, it is essential to avoid future flood events by unveiling critical challenges among complex drivers. Determining the degree of importance and degree of influence of all key flood drivers based on holistic document reviews, interviews, and FGDs sing, this study provides benefits in understanding how cross-border cooperation and regional/national level policies can be implemented to manage flood risk effectively.

Result of MICMAC analysis, it is found the spatial plan, and the stakeholder cooperation and coordination are key flood risk drivers in the transboundary management perspectives. While the former one brings issue in the lack of development control, the latter one carries issue in the failure of transboundary river governance arrangement. "How to strengthen development control in the Ciliwung river basin?" and "What suitable form for transboundary river governance in Ciliwung river basin?". Concentrated efforts to address both challenges are paramount above others to reducing the flood risk.

Finally, to accelerate Ciliwung flood risk reduction, this study suggests a formulation of national policy regarding development control in the Ciliwung river basin and establishment of the Ciliwung river basin commission to improve governance of transboundary river flood management. Furthermore, to formulate an in-depth strategy, future research to deeply investigate development control problems in each region and transboundary institutions' interaction in the Ciliwung River Basin management are required.

To conclude, the research findings on the key flood drivers is very important. They may influence the entire flood risk management system. A proposal is given for a cross-border river basin governance model that can be adapted for flood risk management in other areas with similar characteristics.

**Data Availability Statement**

All data, models, or codes that support the findings of this study are available from the corresponding author upon reasonable request.

**Declaration of competing interest**

The authors declare that they have no known competing financial interests or personal relationships that could have appeared to influence the work reported in this paper.

**Acknowledgment**

This paper is developed as a part of the project Mitigating Hydrometeorological Hazard Impacts Through Improved Transboundary River Management in The Ciliwung River Basin, particularly for Working Package 4 Transboundary Governance Arrangement in Ciliwung River Basin. The project is funded by Indonesia's Ministry of Research and Technology/National Research and Innovation Agency (Kementerian Riset dan Teknologi Republik Indonesia/BRIN) and the UK's Natural Environment Research Council.

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
