# Peer review of "Unveiling Transboundary Challenges in River Flood Risk Management: Learning from the Ciliwung River Basin"

_Natural Hazards and Earth System Sciences, 2023_

## Referee Comment (RC3)

The paper comprehensively discusses the challenges of managing transboundary river flood management along the Ciliwung River Basin, which largely covers the Jakarta Metropolitan Area. While the topic is timely, several issues need to be addressed before considering the paper for publication: (1) cite more recent literature on Greater Jakarta's urbanisation and peri-urbanization/ suburbanisation, which amplify the magnitude of the floodings; (2) add some key/ critical quotes from the expert interviews in the Discussion; (3) make sense the findings by outlining its relevance for or contribution to the global literature development in the areas of transboundary river/ flood risk management; and (4) address grammatical and typo errors throughout the paper (e.g. incomplete or fragmented sentences and paragraphs, non-standard citations, incorrect table numbering/ cross-reference, use of articles).

---

## Author Comment (AC2)

Review

The paper comprehensively discusses the challenges of managing transboundary river flood management along the Ciliwung River Basin, which largely covers the Jakarta Metropolitan Area. While the topic is timely, several issues need to be addressed before considering the paper for publication: (1) cite more recent literature on Greater Jakarta's urbanisation and peri-urbanization/ suburbanisation, which amplify the magnitude of the floodings; (2) add some key/ critical quotes from the expert interviews in the Discussion; (3) make sense the findings by outlining its relevance for or contribution to the global literature development in the areas of transboundary river/ flood risk management; and (4) address grammatical and typo errors throughout the paper (e.g. incomplete or fragmented sentences and paragraphs, non-standard citations, incorrect table numbering/ cross-reference, use of articles).

| No | Reviewer #2 | Harkunti P. Rahayu Response |
|---|---|---|
| 1 | Cite more recent literature on Greater Jakarta's **urbanization** and **peri-urbanization/ suburbanization**, which amplify the magnitude of the floodings; | **Add to after Line 40:**

Several factors have contributed to the increased magnitude of flood impacts in Greater Jakarta over the past few decades (Budiyono et al., 2016), i.e. precipitations, land-use, sea level rise and land subsidence. Meanwhile urbanization in Jakarta between 1995-2014 decreased runoff regulation, which related to a decrease in green spaces and bodies of water, as well as landscape pattern changes (Maheng et al, 2021), and spatial characteristics, which included industrial parks, mixed-use new towns and large-scale residential areas, and shopping centers (Firman and Fahmi, 2017).

However, massive urbanization in Jakarta has strongly influenced the land use change. Silver (2007) has identified that the Jakarta land use change was initiated in the early 1980s, where many agricultures and forest area in suburban of Jakarta were transformed into large-scale subdivisions and new towns. There were more than 30 large new suburban towns and industrial parks built in the peripheries of Greater Jakarta between 1990 and 2010, with average size from 500 to 30,000 hectares (Firman, 2014).

As Jakarta lies in lowland area with 13 river system. All tributaries and river basins located in the Greater Jakarta and its peripheries. The vast peri-urban development in the last three decades may cause the massive conversion of water catchment area, wetland, and green areas, which lead to the increase of flood threat to the Jakarta.

Moreover, Priyambodo et al (2022) study found that urbanization was clearly seen to increase not only the intensity and volume of precipitations, but also the runoff, river flow discharges, which all lead to the increase of flood threat. |
| 2 | Add some key/ critical quotes from the expert interviews in the Discussion | 1. **Spatial plan:** quote will be added to Line 215-229:

[revised manuscript text omitted]

Findings of this study could be as the inputs for other cities/metropolitan which has similar problem transboundary river flood management. |
| 4 | address grammatical and typo errors throughout the paper (e.g. incomplete or fragmented sentences and paragraphs, non-standard citations, incorrect table numbering/ cross-reference, use of articles). | a.  Grammatical and typo errors (incomplete or fragmented sentences and paragraphs) have been check using Grammarly Check Tool and proofread.
b.  Non standard citations have been checked and revised.
c.  Incorrect table numbering has been revised.
d.  Incorrect cross-reference has been revised.
e.  Uses of articles have been check using Grammarly Check Tool and proofread. |

---

## Author Response (AR1)

**Unveiling Transboundary Challenges in The Ciliwung River Flood Management**

The authors wish to thank the editors for their time in effort in reviewing our manuscript. We hope the changes listed have made the manuscript suitable for publication and we look forward to your response.

| No | Editor Comment | Response |
|----|----------------|----------|
| 1. | **Novelty of the Study**: While your research presents an intriguing case study, its novelty remains unclear. It's essential to articulate what makes your work unique, both at the *outset* and *within the discussion*. Ask yourself, | |
| | a. Why should someone outside my research area find this study valuable? | a. This research provides benefits in understanding what are the most important key flood drivers from the perspective of transboundary river management. This study addresses broader issues of key flood drivers in transboundary flood risk management and how transboundary collaboration in regional/national level policies can be implemented to manage flood risk effectively. Findings from this study contribute to several related research themes i.e., sustainable development, disaster risk reduction, environmental science, and transboundary governance.

Note: These paragraph has been adopted in the revision of Introduction section. |
| | b. If you were to present your findings to a global audience, what would they glean from your work? | b. Global audiences can get several benefits. First, the study identified flood risk drivers and highlighted the key flood drivers as part of a comprehensive flood risk reduction strategy. Second, the use of MICMAC analysis as a tool to identify the driving power and dependency power of key flood drivers can be replicated as an innovative approach to assess key flood drivers in another transboundary river basin. The paper would give an advantage in the importance of implementing a transboundary approach in flood management, which can be applied in various regions that face similar challenges.

Note: These paragraph has been adopted in the revision of Result and Discussion as well as in the Conclusion sections. |
| | c. Are there insights they can draw from your methods and their application? | c. The methodological approach in this research, especially the use of MICMAC analysis, can be an alternative method to be applied in flood risk management research contexts. Researchers and policymakers can |

| No | Editor Comment | Response |
|---|---|---|
| | | apply this analysis to identify key flood drivers of complex problems, in this case, related to flood management. This structured approach can result in more informed decision-making in flood risk management.

Note: These paragraph has been adopted in the revision of Result and Discussion section. |
| | d. What is the novel contribution, and how can it benefit them? | d. Novel contribution:
The Jakarta Metropolitan Area floods are complex, many key factors become flood risk drivers. O'Donnel and Thorne (2020) identified several drivers and grouped them into source, pathway, and receptor. Of the many key flood drivers, it is necessary to determine the degree of importance and degree of influence of all flood drivers. Thus, there is still limited research that mentions the most important key flood driver. This research findings on these key flood drivers may influence the entire flood risk management system. From the findings, a proposal is given for cross-border river basin governance for flood management in other area with similar characteristics.

Note: These paragraph has been adopted in the revision of Conclusion section. |
| 2. | **Broader Context**: The manuscript currently lacks a connection to the broader literature and works of other researchers. It's crucial to establish.

a. where your study fits within this larger context. | a. This study of transboundary flood risk management in the Ciliwung River Basin is placed in a broader context, i.e., disaster management, environmental science, and governance. This is particularly relevant for areas of research involving the management of shared water resources, the impact of regional development on flood risk, and strategies to reduce economic losses from flooding. With a research emphasis on the transboundary administrative characteristics of the Ciliwung River basin, this research offers a unique perspective on the challenges and solutions associated with flood management involving several administrative areas.
Note: These paragraph has been adopted in the revision of Introduction section. |

| No | Editor Comment | Response |
|---|---|---|
| | b. how it relates to existing knowledge. | b. This research began with the desk study of key flood drivers from the literature which was verified with the results of interviews in the field. Although previous research has examined flood risk drivers separately, this research takes a systematic approach and focuses on all factors to understand the relationship among all factors. The identification of the most important driving factors adds to the literature by highlighting specific areas that require attention in managing transboundary flood risks. This study contributes to a deeper understanding of the complexities involved in managing flood risk in transboundary river management.

Note: These paragraph has been adopted in the revision of Introduction section. |
| 3. | **Technical Definitions**: In both the Abstract and Introduction sections, you've used the acronym "MICMAC analysis" without offering an immediate definition. It's advisable to introduce and explain any technical term upon its first mention, ensuring readers have clarity from the start. | We added brief technical definition of MICMAC in line 90. Then details in the Methodology Section. |
| 4. | **Results Section**: The current presentation of your results is insufficient. Although you've detailed four methodological steps, the results section only displays a figure. This section should be more expansive, delving into the outcomes of each step. Considering its brevity, you might consider integrating the results with the discussion | We combine the results and discussion section to show a more comprehensive discussion on research finding. |
| 5. | **Placement of Recommendations**: Including recommendations within the discussion section seems out of place. Instead, it may be more effective to weave a succinct version of the recommendations into the conclusion. | We combine the recommendation into the conclusion section. |
| 6. | **Reference Accuracy**: There appear to be inconsistencies and errors in your citations. For instance, in the recommendations section, you cite "(Neuvel & van den Brink, 2009)." and "(Budiyono et al., 2016)" without full integration into the text. Please review and ensure all references are appropriately cited and formatted. | Done for all the referencing. |

---

## Author Response (AR2)

**Harkunti P. Rahayu Response to Animesh Gain, Executive Editor, NHESS Final Comment per 26 November 2023.**

| Animesh Gain, Executive Editor, NHESS Comment | Harkunti P. Rahayu Response |
|---|---|
| I appreciate your submission of the revised manuscript titled "Unveiling Transboundary Challenges in The Ciliwung River Flood Management" (NHESS-2023-85) to our journal. | Thank you very much |
| Upon review, I acknowledge the improvements made over the previous version. However, there remain significant areas that require further revision. Although currently classified as a 'minor revision', please be advised that without addressing these key issues, the manuscript may not be accepted for publication. In this light, I urge you to consider submitting a more substantially revised version. | The suggestion has been accommodated for through review by myself from page 1 to the last page, by having more argument and more literature so the position of the paper in area of flood risk management as well as the transboundary flood risk management. |
| Additionally, for a clearer comparison, please include a document highlighting the changes made from the previous submission to the new version. | Done. |
| The manuscript predominantly reads as a case study. Unfortunately, as it stands, such a format does not align with the publication criteria of NHESS. To move towards acceptance, your manuscript must establish a stronger connection with the broader body of literature and the works of other researchers in this field. It is imperative to identify and articulate the global research gaps your study addresses, not just those pertaining to your specific study area. | The revision has included the better connection with broader body of literature and other works of other researchers. To identify and articulate the global research gaps your study addresses has beedn in the revision. |
| Your manuscript must clearly demonstrate its unique contribution and relevance in a global context. Consider how your findings would be perceived and valued by an international audience. What insights could they derive from your methodologies and their applications? It is crucial to explicitly state the novel aspects of your work and how these contribute to the broader scientific community. | The revision has considered to demonstrate clear unique contributions and relevance to a global context of flood risk drivers. The findings have been elaborated and discussed more in depth. The novel aspects of the study has been elaborated clearly. |
| I am eager to review the next version of your manuscript, upon which I will base my decision. | Thank you, looking forward to good results. |

---

## Author Response (AR3)

**NHESS-2023-85**

**Manuscript Title: Unveiling Transboundary Challenges in The Ciliwung River Flood Management**

The authors wish to thank the editors and reviewers for their time in effort in reviewing our manuscript. We hope the changes listed have made the manuscript suitable for publication and we look forward to your response.

| No | Comment | Response |
|---|---|---|
| | **Editor** | |
| 1. | The manuscript predominantly reads as a case study. Unfortunately, as it stands, such a format does not align with the publication criteria of NHESS. | We have restructure the paper, introduction cover:
• The issues of transboundary river flood management, the impact of the extension of urban area to become urban agglomeration which worsen the challenges.
• Key research problem is being addressed and the terminology associated with it
• The introduction is then highlight the research problem and aims.
• The benefit of the study for novel contribution in the context of global research. |
| 2 | To move towards acceptance, your manuscript must establish a stronger connection with the broader body of literature and the works of other researchers in this field.

It is imperative to identify and articulate the global research gaps your study addresses, not just those pertaining to your specific study area. You should specify your novel contribution in the context of global research, not just based on your case study. | We have added more in depth literature review.

We have widened the angle of research scope, not only identifying the transboundary flood challenges/drivers but also prioritizing the drivers to be discussed for contributing to transboundary governance. |

| No | Comment | Response |
|---|---|---|
| 3 | | There is additional figure to support the description and analysis of the case study for lesson learned. |
| 4 | Your manuscript must clearly demonstrate its unique contribution and relevance in a global context. Consider how your findings would be perceived and valued by an international audience. What insights could they derive from your methodologies and their applications? It is crucial to explicitly state the novel aspects of your work and how these contribute to the broader scientific community. | We have revised the paper as suggested.
The revised paper has also been reviewed by the co-author Richard Haigh, followed by some major changing. |